# Application of Taguchi Method to Optimize the Parameter of Fused Deposition Modeling (FDM) Using Oil Palm Fiber Reinforced Thermoplastic Composites

**DOI:** 10.3390/polym14112140

**Published:** 2022-05-24

**Authors:** Mohd Nazri Ahmad, Mohamad Ridzwan Ishak, Mastura Mohammad Taha, Faizal Mustapha, Zulkiflle Leman, Debby Dyne Anak Lukista, Ihwan Ghazali

**Affiliations:** 1Department of Aerospace Engineering, Faculty of Engineering, Universiti Putra Malaysia, Serdang 43400, Selangor, Malaysia; faizalms@upm.edu.my; 2Faculty of Mechanical and Manufacturing Engineering Technology, Universiti Teknikal Malaysia Melaka, Hang Tuah Jaya, Durian Tunggal 76100, Melaka, Malaysia; mastura.taha@utem.edu.my (M.M.T.); debbydyne2024@gmail.com (D.D.A.L.); ihwan@utem.edu.my (I.G.); 3Centre of Smart System and Innovative Design, Universiti Teknikal Malaysia Melaka, Hang Tuah Jaya, Durian Tunggal 76100, Melaka, Malaysia; 4Aerospace Malaysia Research Centre (AMRC), Universiti Putra Malaysia, Serdang 43400, Selangor, Malaysia; 5Laboratory of Biocomposite Technology, Institute of Tropical Forestry and Forest Products (INTROP), Universiti Putra Malaysia, Serdang 43400, Selangor, Malaysia; 6Department of Mechanical and Manufacturing Engineering, Faculty of Engineering, Universiti Putra Malaysia, Serdang 43400, Selangor, Malaysia; zleman@upm.edu.my; 7Advanced Engineering Materials and Composites Research Centre, Faculty of Engineering, Universiti Putra Malaysia, Serdang 43400, Selangor, Malaysia; 8Department-General Education, Faculty of Resilience, Rabdan Academy, Abu Dhabi 22401, United Arab Emirates

**Keywords:** FDM, Taguchi, DOE, oil palm fiber composites, ABS

## Abstract

Fused Deposition Modeling (FDM) is capable of producing complicated geometries and a variety of thermoplastic or composite products. Thus, it is critical to carry out the relationship between the process parameters, the finished part’s quality, and the part’s mechanical performance. In this study, the optimum printing parameters of FDM using oil palm fiber reinforced thermoplastic composites were investigated. The layer thickness, orientation, infill density, and printing speed were selected as optimization parameters. The mechanical properties of printed specimens were examined using tensile and flexural tests. The experiments were designed using a Taguchi experimental design using a L_9_ orthogonal array with four factors, and three levels. Analysis of variance (ANOVA) was used to determine the significant parameter or factor that influences the responses, including tensile strength, Young’s modulus, and flexural strength. The fractured surface of printed parts was investigate using scanning electron microscopy (SEM). The results show the tensile strength of the printed specimens ranged from 0.95 to 35.38 MPa, the Young’s modulus from 0.11 to 1.88 GPa, and the flexural strength from 2.50 to 31.98 MPa. In addition, build orientation had the largest influence on tensile strength, Young’s modulus, and flexural strength. The optimum printing parameter for FDM using oil palm fiber composite was 0.4 mm layer thickness, flat (0 degree) of orientation, 50% infill density, and 10 mm/s printing speed. The results of SEM images demonstrate that the number of voids seems to be much bigger when the layer thickness is increased, and the flat orientation has a considerable influence on the bead structure becoming tougher. In a nutshell, these findings will be a valuable 3D printing dataset for other researchers who utilize this material.

## 1. Introduction

Additive Manufacturing (AM) is layer by layer-oriented or layered manufacturing, which was first established as rapid prototyping technology towards the end of the 1980s. It delivers minimal cost, adaptable goods; as a result, the application of this technology to build enhanced parts has been a popular trend in recent years [1]. Figure 1 depicts a schematic representation of FDM. It is particularly useful in the formation of thermoplastics such as ABS, polylactic acid (PLA), and polypropylene (PP) [2,3]. Anisotropic characteristics were found on an ABS specimen created using FDM [4]. The quantitative and qualitative properties of thermoplastic specimens are determined by the experimental optimization of the FDM input parameters. The proper setup of the process parameters helped to provide a consistent and accurate finished ABS product, enhance process sustainability, and decrease post-processing activity [5,6,7]. Layer thickness, raster width, orientation, and infill density percentage all had an impact on the mechanical characteristics of thermoplastic when processed with FDM [8]. The use of thermoplastic cellulose-based composite filaments in FDM is exciting, since it reduces raw material cost [9,10] and has good impact on the environment [11,12,13,14], reduces warping after processing [15], and preserves the material’s mechanical attributes [16].

Recently, many researchers employed natural fiber reinforced composite (NFRC) materials such as PLA/pineapple [17], iron/nylon [18], ABS/fiberglass [19], ABS/carbon fibers [1], and PLA/wood [20] to create specimens for mechanical testing using FDM. All of these factors should be optimized and controlled efficiently in FDM to produce high dimensional accuracy, quality of printed parts, less distortion, reduced porosity, and superior mechanical properties [21]. Many researchers had concentrated on the optimization of the machine’s process parameters and were concerned with understanding the link between different processing factors and their impacts on the final product. Accordingly, the study mostly focused on characteristics such as ambient temperature [22], layer thickness [23], extrusion temperature [24], printing speed [25], reinforced particle size [26], orientation of raster [27], road width [28], and raster air gap [29]. Huynh et al. [30] investigated the impact of printing speed, pattern, and layer thickness on the dimensional accuracy of the printed parts. Meanwhile, Raut et al. [31] investigated the mechanical properties and the processing time of printed specimens. El-Kassas and Elsheikh [32] investigated the rice straw fiber composite utilizing a novel approach that offers several benefits over traditional procedures, including the absence of chemicals or heat treatments, a minimum manufacturing area, and lower production costs.

The solution to optimize the FDM input parameters involved applying several design of experiments (DOE), including ANOVA, fuzzy logic, full factorial design, response surface methodology (RSM), and the Taguchi method [33]. Because of its robustness, tolerance, and dimensional control, Taguchi’s design of experiments approach was commonly used to optimize the FDM parameters setting [34]. Ahmad et al. [35] used the Taguchi approach to improve surface roughness of the printed parts by optimizing the printing parameters of FDM. On the other hand, the Taguchi approach is a descriptive survey that identifies a product or process in order to enhance its usability. The Taguchi technique is employed since it is simple and a problem-solving technique to help improve process performance, to increase efficiency and productivity. The approach is known as the factorial outline of the test. The orthogonal technique is used to select the level combination of information plan variables for each experiment [36]. Taguchi’s major goal is to optimize a process’s parameters to attain maximum efficiency [37]. The Taguchi technique was used to optimize the FDM input parameters such as a printing pattern, printing speed, layer thickness and orientation of raster. The results demonstrated that layer thickness had a substantial impact on Young’s modulus at a 20° displacement angle [38,39]. The lowest layer thickness improves tensile strength while also conserving raw material [40]. Kumar et al. [41] investigated the effect of five factors on the surface integrity of ABS parts manufactured using FDM. To understand the influence of the treated parameter, he employed Taguchi analysis and did analysis of variance (ANOVA). ANOVA also was used to identify the significant factor contributing to the experimental conditions [42].

The use of NFRC in FDM is impressive since it reduces cost of feedstock material and has a low environmental effect [9,10]. It also reduces warping after processing though perhaps keeping the material’s mechanical strengths [16]. However, production process becomes problematic owing to issues shared with typical cellulose-based composites, such as feedstock drying non-uniformity in filler distribution, void formation, and temperature control [43,44]. Recently, several researchers have been focusing on the creation of novel materials to increase mechanical performance. There has not been enough study done to discover the process parameter interdependence on various materials in order to enhance the mechanical characteristics of printed parts by FDM. Additionally, no previous research has investigated the optimal parameter settings for FDM to print products manufactured from oil palm fiber reinforced thermoplastic composites. Thus, it is vital to investigate the optimum printing parameters for new materials like oil palm fiber composite in order to get the best results and to provide primary data to other researchers that use this material.

## 2. Materials and Methods

### 2.1. Materials

The oil palm fibers or known as *Elaeis guineensis* used in this research were obtained from a local estate. Meanwhile, the ABS types PA-747H were supplied by a local supplier that is imported from the Chi Mei Corporation. The material preparation process of oil palm fiber composite in granules form include the method of fiber treatment, mixing, hot pressing, crushing, as was explained in our previous study [45]. The oil palm fiber composite filament was fabricated using a twin screw extruder. The extruder was built locally in Malaysia, using a Siemens PLC controller that was imported from China. Figure 2 shows the process flow of this study for fabrication and optimization of oil palm fiber reinforced thermoplastic composites. The method of research starts with the material pre-processing of the fiber and the compounding process. Then, it continues with the fabrication of the filament by an extrusion process, printing specimens, application of DOE, performing mechanical test, and lastly evaluating the fractured surface of specimens by SEM. For this study, 3wt% of oil palm fiber was used to study the printing optimization using an FDM 3D printer.

### 2.2. Design of Experiment (DOE)

The relation between the experimental factors and their response was evaluated using the Taguchi method. The design of experiments was carried out using Minitab-16 software to examine the effect on mechanical strength of the oil palm fiber composite specimens. Orthogonal array design L_9_ was generated by Minitab software by referring to the four factors and three levels. It is selected in this study because it contains a minimum number of experiments so that it becomes more affective, reducing experiment time and cost. Figure 3 shows the graph of tensile strength, Young’s modulus, and the flexural strength based on the nine design runs. According to the literature, the strength of composites is determined by numerous characteristics such as nozzle temperature, raster orientation, raster angle, contour width, infill density, orientation, printing speed, layer height, and thermal conductivity. There were four key characteristics that determine the strength of the printed specimens that were addressed in this investigation, including layer thickness, orientation, infill density, and printing speed, as shown in Table 1. The tests were carried out according to the design matrix, and the results are also reported. Table 2 shows the number of experiments for orthogonal array L_9_ (3^4^) as well as the response values for nine design runs. The Taguchi approach includes using a rigorous experimental design to optimize process parameters. Tensile strength, Young’s modulus, and flexural strength were chosen as responses to calculate the percentage contribution of each factor. The selection of levels of each factor is based on the minimum, medium and maximum range of printing parameters using an FDM 3D printer. The signal-to-noise (S/N) ratio is a quality metric for assessing the impact of input variables towards responses. All of the output responses in this investigation are quality characteristics of the ‘larger the better’ type. Equation (1) was used to estimate the *S/N* ratios for this characteristic,
(1)$$S/N=-10\;\log_{10}\left(\overset{n}{\underset{i=1}{\sum }}Y_{i^{2}}\right)$$
where *S/N* is signal-to-noise; *i* is 1, 2, 3, … *n*; *Y* is output response value.

### 2.3. 3D Printing

A 3D printer, model FlashForge, Creator Pro (Zhejiang Flashforge 3D Technology Co., Ltd., Jinhua City, China) was used to print all specimens. The selected printing parameters of FDM to be optimized are orientation, infill density, layer thickness, and printing speed, as shown in Figure 4a–d. The printing orientations were 0, 45, and 90°, with layer thicknesses (0.2, 0.3, and 0.4 mm), infill density (0, 50 and 100%) and printing speed (10, 50, 100 mm/s), respectively. A FlashPrint, version 5.3.1 software (Zhejiang Flashforge 3D Technology Co., Ltd., Jinhua City, China) was used to integrate with 3D printer. It provides a simple and easy to use user interface for preparing of 3D designs for printing on the FlashForge 3D printers. The tensile and flexural samples were printed according to ASTM 638 and ASTM 790 standards.

### 2.4. Tensile Testing

The tensile properties of the samples were analyzed following a standard procedure of ASTM D-638 and the dimension of printed specimens can be referred to Table 3. The determinations of tensile strength and modulus as well as elongation were performed in a universal testing machine (Shimadzu Autograph AGSX, Shimadzu Corporation, Kyoto, Japan) with a 50 kN load cell and constant crosshead speed of 5 mm/min. Tensile strength is a measurement of a material’s ability to withstand stretching or the extent to which it can be stretched before failing. The composites’ tensile characteristics, such as stiffness, ductile modulus, and prolongation at breaking point, were measured. The formulae for calculating tensile strength and modulus of elasticity are Equations (2) and (3).
(2)σ=F/A
where  σ is the tensile strength at yield; *F* is the force applied (kN); *A* is the cross-section area in rectangular.
(3)E=σ/ϵ
where *E* is the young’s modulus (MPa); σ is the stress applied on printed specimen; ϵ is the strain of the specimen (mm/mm).

### 2.5. Flexural Testing

A flexural test, also known as a three-point bending test, was used to measure the bending strength and modulus of printed specimens made from ABS-oil palm fiber composite using FDM. The flexural properties of the samples were analyzed according to a standard procedure of ASTM D-790 and the dimension of printed specimens as in Table 3. A universal testing machine (Shimadzu Autograph AGSX, Shimadzu Corporation, Kyoto, Japan) equipped with a 50 kN load cell was used to conduct flexural tests at 23 ± 1 °C temperature and of 50 ± 5% relative humidity. Yield and fracture parameters were evaluated at a crosshead speed of 5 mm/min. The ability of a material to tolerate twisting in the opposite direction of its axial center is known as flexural ability. Furthermore, when a bar-shaped test piece with a straight shaft is exposed to a twisting force opposed to the bar, the maximum pressure is formed [46]. Two support beams were used to hold the printed specimens. The specimens were then loaded using a loading nose in the middle. The span-to-depth ratio (*R*) of the structure was 16:1. Equation (4) was used to compute the rate of crosshead motion for each specimen. Equation (5) was used to compute the flexural stress from the observed load, whereas Equation (6) was used to determine the flexural modulus.
(4)$$R=\frac{ZL^{2}}{6d}$$
(5)$$\sigma_{max}=\frac{3PL}{2bd^{2}}$$
(6)$$E_{H}=\frac{L^{2}P}{4bd^{3}}$$
where *E_H_* is the flexural modulus (MPa); *L* is the support span (mm); *P* is the load at yield; *σ_max_* is the flexural strength (MPa); *d* is the thickness (mm); *b* is the width (mm) (a^2^ + b^2^ = c^2^); *R* is the rate of crosshead motion, mm/min; *Z* is the rate of straining of the outer fiber, mm/min; and *Z* is 0.01.

### 2.6. Scanning Electron Microscope (SEM)

The morphology of the fractured surface of ABS-oil palm fiber composite was observed using a scanning electron microscope, model ZEISS LEO 1455 VPSEM (Jena, Germany) at 10 kV acceleration voltage. All samples were cut to a standard size and platinum-coated on the surface prior to the experiment. All test samples were stored in zip plastic bags after inspection.

## 3. Results and Discussion

### 3.1. Interaction Plots, Probability Plots and ANOVA

The interaction plots for tensile strength, Young’s modulus, and flexural strength as a function of four parameters considered in this study are shown in Figure 5. In this interaction plot, the lines were not parallel, which indicates the good relationship among the factors to the value of tensile strength, Young’s modulus, and flexural strength. However, there was no interaction between the orientation-infill and the orientation-printing speed towards mean of tensile strength and Young’s modulus. Parameter setting for 0.2 mm layer thickness, 0° of build orientation, 100% infill density indicate the 10 mm/s speed gave the higher tensile strength, Young’s modulus and flexural strength. Irfan et al. [47] found that there was no interaction between fiber mass and feeding zone to the tensile strength of kenaf-polypropylene composites.

Probability plots quantify the dispersion of experimental results of tensile strength, Young’s modulus, and flexural test of oil palm fiber composite specimens. Normal probability plots are utilized to determine if data comes from a normal distribution. The statistical technique is based on the assumption of a normal underlying distribution [48]. As a result, normal probability plots can either give comfort that the assumption is justified or provide a warning that the assumption has flaws. Normal probability plots and hypothesis tests for normalcy are usually combined in a normality study. An assumption of normality is plausible in a normal probability plot if all of the data points lie around the line. Otherwise, the points will curve away from the line, eliminating the need for a normalcy assumption. The Anderson Darling (AD) test is used to validate the normalcy assumption [49]. It is a sophisticated statistical technique used to discover outliers from normality. Figure 6 indicates that the experimental data for all answers is close to the fitted line, the AD statics values are low, and the *p*-value of the test is larger than 0.05, implying that the data follows a normal distribution. As a result, the data may be subjected to additional analysis and optimization. The normal probability plots show that all points are close to a straight line and are evenly distributed, with no outliers. On the other hand, it means that the errors are distributed normally. This pattern also shows that residuals are evenly distributed throughout all runs. Montgomery [50] used ANOVA to demonstrate how the two factors influence Young’s modulus and tensile strength.

To investigate the primary influence of input parameters on individual response, an ANOVA with a 95% confidence interval was used. The ANOVA values for tensile strength, Young’s modulus, and flexural strength are shown in Table 4. If the *p*-value is less than 0.05, the parameter is significant in terms of responses. The most significant factor influencing the value of tensile strength in this investigation was orientation, which contributed 67.7%. The remaining factors, such as layer thickness, printing speed, and infill density, were not significant according to *p*-values larger than 0.05. The greater influence parameter for Young’s modulus response was also orientation, with a 45.5% contribution and a *p*-value less than 0.05. Furthermore, other parameters were minor, contributing a lesser percentage of contribution. This also applied to flexural strength value, where part orientation was the most influential parameter, accounting for 49.6% of the response. This finding was in agreement with Chacón et al. [51], who had observed the effect of layer thickness, built orientation, and feed rate on the strength of the printed parts by FDM. He found that the 0° of build orientation was the influencing parameter as relates to the value of tensile strength.

### 3.2. Main Effect Plots for Means

A main effects plot is a graph that displays the mean response values for each level of a design parameter of FDM. It is useful for comparing the relative strength of several factors’ impacts. Figure 7a–c show the main effect plot of means for tensile strength, Young’s modulus, and flexural strength, respectively. The result shows that the tensile strength increases when layer thickness, infill density and printing speed increase from low to high level. However, when the orientation parameter increases from low to high, the value of tensile strength decreases. The graph for Young’s modulus demonstrates that the value of the modulus rose from level 2 to 3 for layer thickness, speed, and infill density, but not for orientation. Furthermore, the main effect plot of means for flexural strength reveals that when the level is increased from 2 to 3, the mean values for all parameters decrease. Overall, printing orientation (0°) contributes a greater value of tensile strength, Young’s modulus, and flexural strength. This result was consistent with the findings by Chacón et al. [51], who discovered that the 0° of printing orientation was the most contributing parameter to the tensile strength value.

### 3.3. Taguchi Optimization by S/N Ratio

Figure 8 shows the main effects plots for the *S/N* ratios of tensile strength, Young’s modulus, and flexural strength with printing parameter such as layer thickness, orientation, infill density, and printing speed. Since the aim of the study is to maximize the value of responses such as tensile strength, Young’s modulus, and flexural strength, the *S/N* ratio was selected to be ‘larger the better’. The optimal levels are obtained by computing the average values of the *S/N* ratios for each response at each level, and the higher values of the *S/N* ratios show good quality characteristics. Irfan et al. [47] studied the compounding parameters of kenaf fiber composite. They applied the *S/N* ratio to analyze the results as a function of the factors and levels for tensile strength and modulus. Figure 8a,b show the *S/N* ratio for tensile strength and Young’s modulus. Meanwhile, Figure 8c shows the *S/N* ratio for flexural strength and trend posters the decreasing the value of the *S/N* ratio from level 1 to level 3. Overall, the optimum printing parameter for FDM using oil palm fiber composite filament was 0.4 mm layer thickness, flat (0 degree) of orientation, 50% infill density and 10 mm/s printing speed. Among all the factors, build orientation became the most significance parameter towards responses including tensile strength, Young’s modulus, and tensile strength. This result was similar to the research finding on FDM optimization by Chacón et al. [51], Domingo et al. [52], Tymrak et al. [53] and Lanzotti et al. [54]. They found that the build orientation plays the main role in the performance of mechanical properties of the printed parts by FDM. The flat (0°) orientation parameter contributes the higher tensile strength rather than increasing the degree (°) of orientation.

### 3.4. SEM Analysis of Fractured Surfaces

Figure 9 shows SEM images of the specimen’s fractured surfaces. Figure 9a–d indicate the fractured surfaces of S1, S4, S7, and S8 specimens from design experiment runs 1, 4, 7, and 8, respectively. The fractured fiber can clearly be observed on Figure 9b,c. It was caused by the extrusion, compounding, and fiber treatment processes [55,56]. The microstructural changes in all specimens with different parameter settings were seen in the cross section images of the samples. Figure 9d shows a cross section pattern with a 45° printing orientation, 0.4 mm layer thickness, 100 mm/s and 100% infill density. It demonstrates that the interlayer printing beads were misaligned as a result of the increased printing speed. However, Figure 9a with parameter 0° orientation shows the tougher bead structure and looks more ductile when compared to the 45° orientation. Microscopic images reveal the presence of voids between beads, resulting in a reduction in the element’s net section and real density, as well as the existence of a cavity at the interaction of beads [57]. The adhesion force between filaments can be reduced by a type of defect, which is most likely connected to the extrusion temperature. When comparing layer thicknesses of 0.2 mm and 0.4 mm, the number of voids appeared to be significantly higher. This finding is in accordance with that of Tekinalp et al. [58], who discovered that the FDM printed fiber reinforced composites had lower inter-layer porosity but higher inner-layer porosity. As shown in the SEM images, the parameter of thickness 0.4 mm, orientation 0°, infill 0.5, and printing speed 10 mm/s was the optimal configuration, with reduced void and good layer arrangement after applying stress. Figure 9b–d show the fiber pull out on the images of the printed specimens. It is due to the poor bonding between fiber and matrix, and when these fibers pull out, the structure’s strength will decrease. If the interfacial adhesion is good, a significant amount of energy is required.

## 4. Conclusions

The Taguchi approach was used to improve the process parameters of FDM, which use oil palm fiber reinforced thermoplastic composites as feedstock. Through the use of an orthogonal array L_9_ (3^4^) to explore the impact of process factors on variable behavior, four parameters with three levels were investigated: layer thickness, orientation, infill density, and printing speed. According to ANOVA analysis, the most significant input variable is build orientation, with *p*-values less than 0.05 for all responses including tensile, Young’s modulus, and flexural strength. According to the normal probability plots, all points are near to a straight line and distributed evenly, with no outliers. The tensile strength of the composites ranged from 0.95 to 35.38 MPa, while Young’s modulus was 0.11 to 1.88 GPa and the flexural strength was 2.50 to 31.98 MPa. The best printing parameters for FDM using oil palm fiber composite filament based on the S/N ratio were 0.4 mm layer thickness, flat (0°) orientation, 50% infill density, and 10 mm/s printing speed. The microscopic views show the existence of voids between beads, causing a reduction in the element’s net section and actual density, as well as the presence of a cavity at the bead interface. Finally, SEM images revealed that the 0° build orientation has a stronger bead structure and seems more ductile when compared to 45° orientations. As a result, the study would provide experts and researchers with accurate data when dealing with this novel material as a feedstock for FDM.

## Figures and Tables

**Figure 1 polymers-14-02140-f001:**
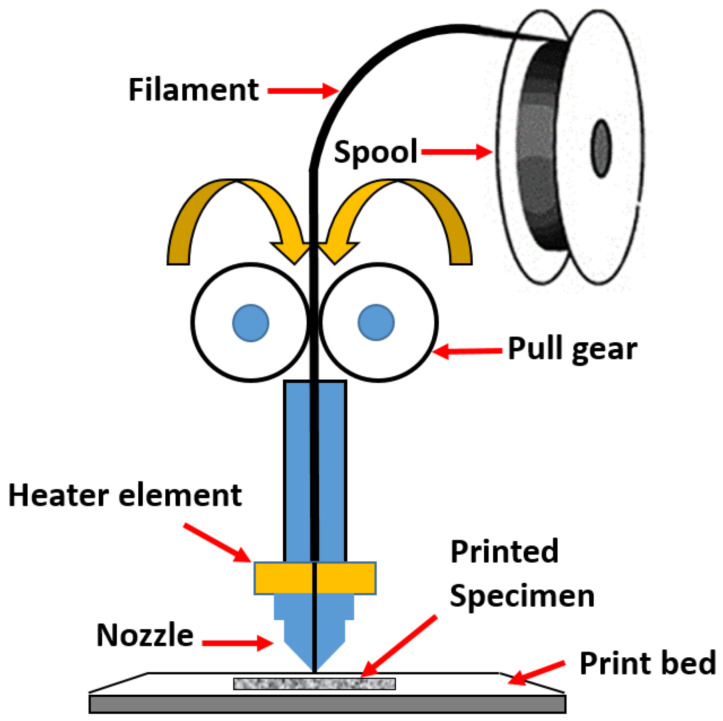
Schematic of a FDM 3D printing.

**Figure 2 polymers-14-02140-f002:**
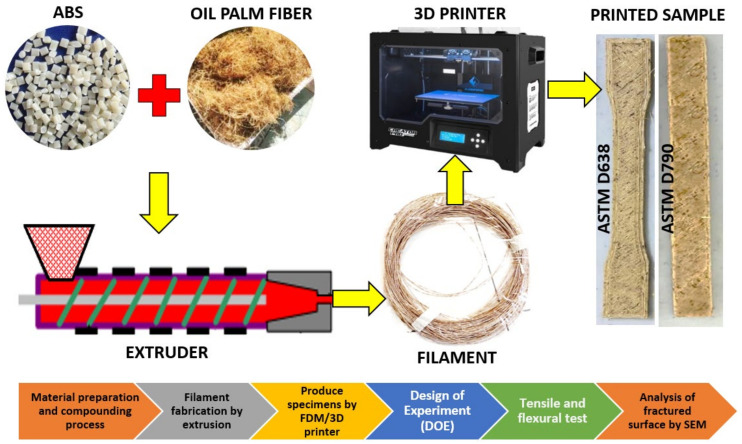
Process flow for fabrication and optimization of oil palm fiber reinforced thermoplastic composites.

**Figure 3 polymers-14-02140-f003:**
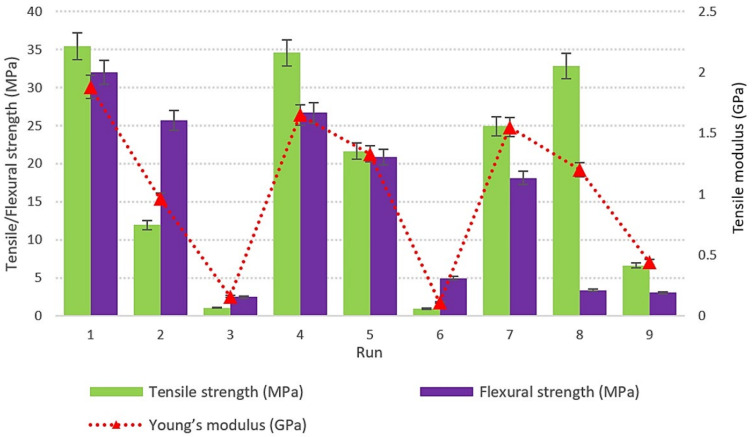
Graph of tensile strength, Young’s modulus and flexural strength of oil palm fiber reinforced thermoplastic composites.

**Figure 4 polymers-14-02140-f004:**
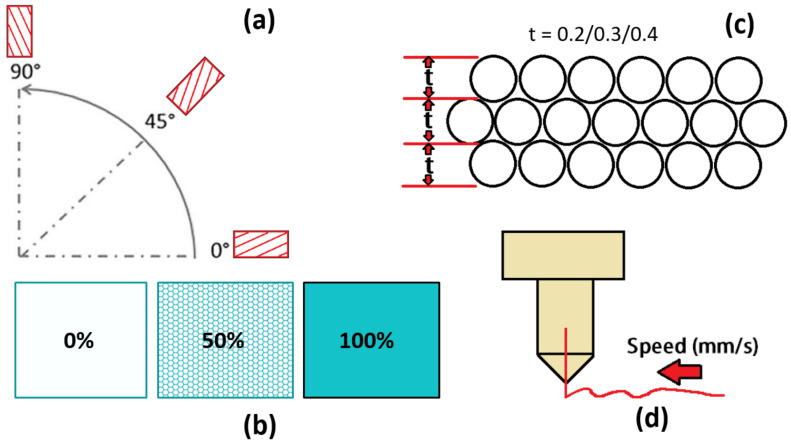
Printing parameter (**a**) orientation (**b**) infill density percentage (**c**) layer thickness and (**d**) printing speed.

**Figure 5 polymers-14-02140-f005:**
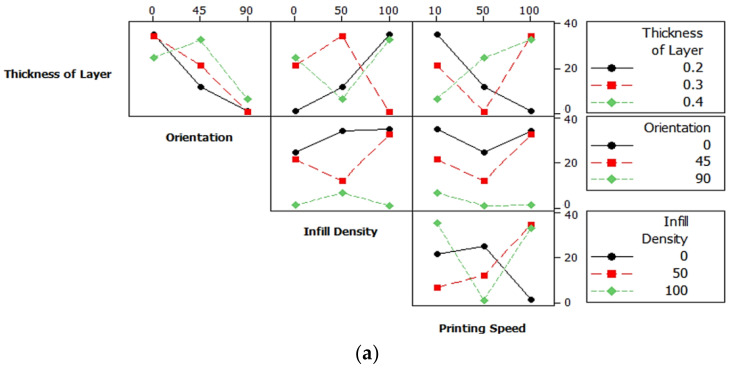
Interaction plots of (**a**) tensile strength (**b**) Young’s modulus and (**c**) flexural strength.

**Figure 6 polymers-14-02140-f006:**
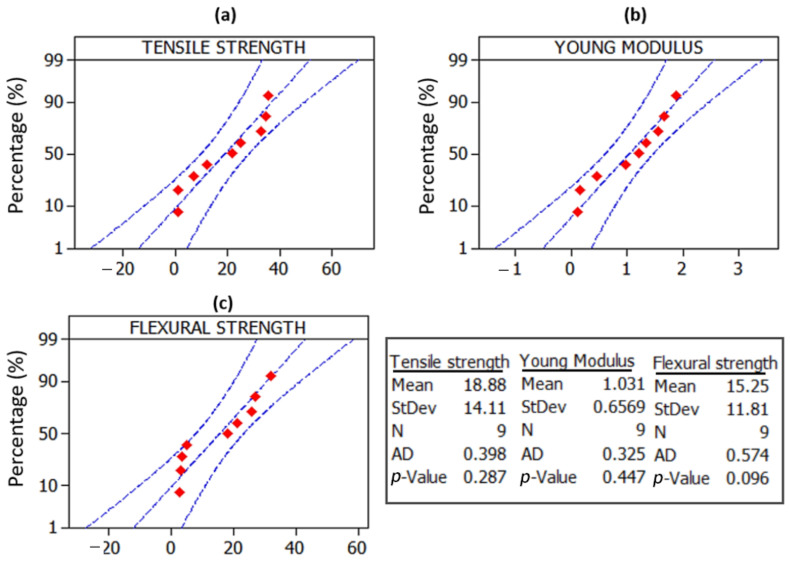
Normal probability plots of (**a**) tensile strength (**b**) Young’s modulus and (**c**) flexural strength.

**Figure 7 polymers-14-02140-f007:**
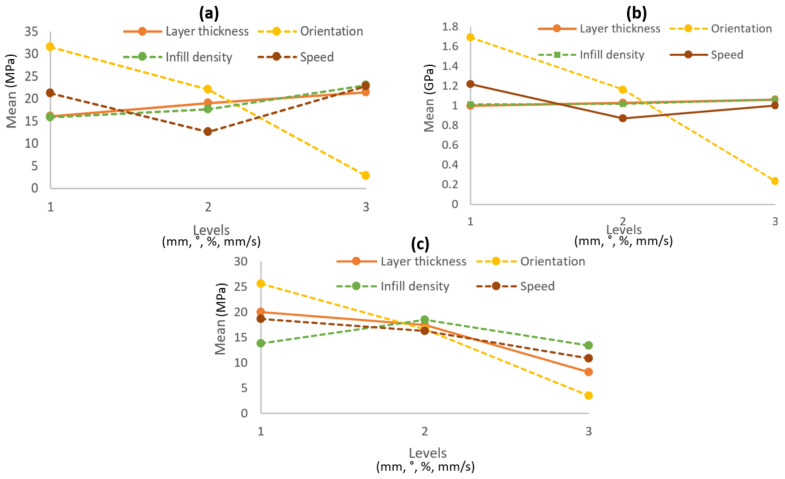
Main effect plot of means for (**a**) tensile strength (**b**) Young’s modulus and (**c**) flexural strength.

**Figure 8 polymers-14-02140-f008:**
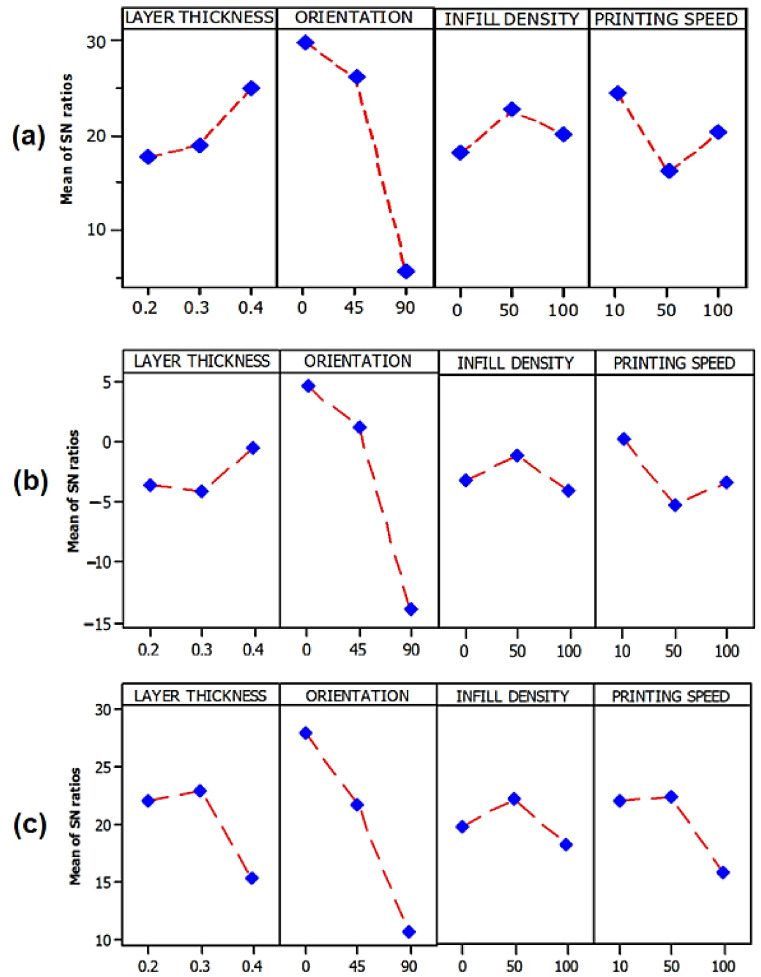
Main effects plot for the *S/N* ratios (larger the better) of (**a**) tensile strength (**b**) Young’s modulus and (**c**) flexural strength.

**Figure 9 polymers-14-02140-f009:**
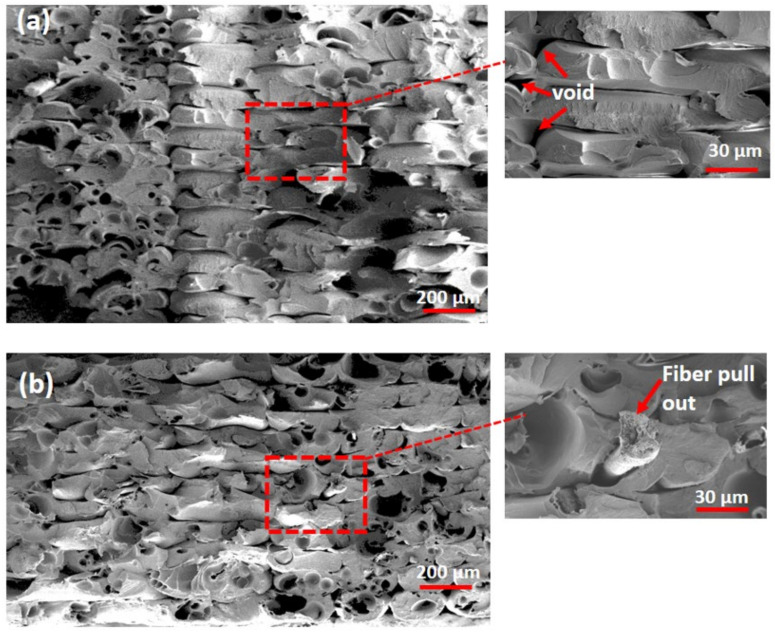
SEM images of (**a**) S1 (**b**) S4 (**c**) S7 and (**d**) S8 specimens with fractured surfaces.

**Table 1 polymers-14-02140-t001:** Factors and their levels.

Factors		Levels		
		1	2	3
A	Thickness of Layer (mm)	0.2	0.3	0.4
B	Orientation on Z-axis (°)	0	45	90
C	Infill Density (%)	100	50	0
D	Printing Speed (mm/s)	10	50	100

**Table 2 polymers-14-02140-t002:** Orthogonal array design L_9_ (3^4^).

Run	Layer Thickness (mm)	Orientation (°)	Infill Density (%)	Printing Speed (mm/s)	Responses		
					Tensile Strength (MPa)	Young’s Modulus (GPa)	Flexural Strength (MPa)
1	0.2	0	100	10	35.38	1.88	31.98
2	0.2	45	50	50	11.94	0.96	25.67
3	0.2	90	0	100	1.06	0.16	2.50
4	0.3	0	50	100	34.55	1.65	26.67
5	0.3	45	0	10	21.64	1.33	20.88
6	0.3	90	100	50	0.95	0.11	4.97
7	0.4	0	0	50	24.92	1.55	18.12
8	0.4	45	100	100	32.83	1.20	3.40
9	0.4	90	50	10	6.64	0.44	3.04

**Table 3 polymers-14-02140-t003:** Dimension of ASTM D638 and ASTM D790 standard specimen.

Dimension (mm)	ASTM D638	ASTM D790
Overall length	165	130
Overall width	19	13
Distance between grips	115	-
Gage length	50	-
Length of narrow section	57	-
Thickness	3.2	3
Radius of fillet	76	-
Gage width	13	-

**Table 4 polymers-14-02140-t004:** ANOVA for each response.

Response	Source	DoF	Adj SS	Adj MS	*F*-Value	*p*-Value	% Contribution	Remarks
Tensile strength	Layer thickness	2	43.0	21.0	0.08	0.921	2.26	Insignificant
	Orientation	2	1286.2	643.1	12.55	0.007 *	67.67	Significant
	Infill density	2	83.0	42.0	0.17	0.851	4.37	Insignificant
	Printing speed	2	181.0	91.0	0.38	0.696	9.52	Insignificant
	Error	6	307.4	51.2			16.17	
	Total	8	1900.6				100	
Young’s modulus	Layer thickness	2	0.006	0.003	0.01	0.995	0.09	Insignificant
	Orientation	2	3.262	1.631	51.23	0.000 *	48.51	Significant
	Infill density	2	0.005	0.002	0.00	0.996	0.07	Insignificant
	Printing speed	2	0.180	0.090	0.17	0.851	2.68	Insignificant
	Error	6	3.272	1.726			48.65	
	Total	8	6.725				100	
Flexural strength	Layer thickness	2	234.0	117.0	0.80	0.494	15.68	Insignificant
	Orientation	2	740.6	370.3	5.91	0.038 *	49.62	Significant
	Infill density	2	47.0	23.0	0.13	0.880	3.15	Insignificant
	Printing speed	2	95.0	48.0	0.28	0.765	6.36	Insignificant
	Error	6	376.0				25.19	
	Total	8					100	

***** Source of variance with *p*-value less than 0.05 is significant; DoF is degree of freedom; Adj SS is the adjusted sum of square; Adj MS is the adjusted mean square.

## Data Availability

The data presented in this study are available on request from the corresponding author.

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
