# Peer review of "Application of Taguchi Method to Optimize the Parameter of Fused Deposition Modeling (FDM) Using Oil Palm Fiber Reinforced Thermoplastic Composites"

_polymers, 2022, doi:10.3390/polym14112140_

Round 1

Reviewer 1 Report

The article is well-drafted. The authors designed the experiment using Taguchi method to optimize the parameters of FDM 3D printing such as layer thickness, orientation, infill density, and printing speed. Some minor changes are needed before being accepted. 

1) The authors used oil palm fiber reinforced thermoplastic composites. What was the filler loading? 

2) A is the cross-section area in equation (2). How was that calculated? Were voids or infill density being considered when calculating cross section?

Author Response

Refer to attach file.

Reviewer 2 Report

The authors investigated the optimum printing parameters of fused deposition modeling using oil palm fiber reinforced thermo-plastic composites were. The layer thickness, orientation, infill density, and printing speed were selected as optimization parameters. The mechanical properties of printed specimens were examined using tensile and flexural tests. The experiments were designed using a Taguchi experimental design using a L9 orthogonal array with four factors, and three levels. Analysis of variance (ANOVA) was used to determine the significant parameter or factor that influencing the responses including tensile strength, young’s modulus and flexural strength. The fractured surface of printed parts was investigate using scanning electron microscopy (SEM). The results show the tensile strength of the printed specimens ranged from 0.95 to 35.38 MPa, the young's modulus from 0.11 to 1.88 GPa, and the flexural strength from 2.50 to 31.98 MPa. In addition, build orientation had the largest influence on tensile strength, Young's modulus, and flexural strength. The optimum printing parameter for FDM using oil palm fiber composite was 0.4 mm layer thickness, flat of orientation, 50% infill density and 10 mm/s printing speed. The results of SEM images demonstrate that the number of voids seems to be much bigger when the layer thickness is increased, and the flat orientation has a considerable influence on the bead structure becoming tougher. As a nutshell, these findings will be a valuable 3D printing dataset for other researchers that utilize this material.

The manuscript is interested and could be published after major revision.

  1. The paper has a good methodology. However the main aim of the paper should be highlighted.
  2. The list of references is extensive. However, the introduction section should be updated.
  3. The introduction must be updated to contain recent publications from Polymers such as “Bistable Morphing Composites for Energy-Harvesting Applications”.
  4. Support your literature by modeling of drilling process of gfrp composite using a hybrid random vector functional link network/parasitism-predation algorithm; effect of surface preparation on the strength of vibration welded butt joint made from pbt composite; recent progresses in wood-plastic composites: pre-processing treatments, manufacturing techniques, recyclability and eco-friendly assessment; a new eco-friendly mechanical technique for production of rice straw fibers for medium density fiberboards manufacturing and others.
  5. Table 5 should be updated. Please used clear fonts.
  6. “Figure 5 shows the main effects plots for the S/N ratios (larger the better) of tensile strength, young’s modulus and flexural strength with printing parameter such as layer thickness, orientation, infill density and printing speed..” support this claim with more discussion.
  7. Why did the authors consider “larger the better”?
  8. There are TWO figure 5?!
  9. What are the units of Y axis and x axis in the mean effect plots shown in Figure 5?
  10. What are the specifications of the used scanning electron microscope?
  11. Why did the authors use Orthogonal array design L9?
  12. How did the authors select the levels of each factor?

Author Response

Refer to the attached file.

Round 2

Reviewer 2 Report

 Accept in present form